# Type II Transmembrane Serine Proteases as Modulators in Adipose Tissue Phenotype and Function

**DOI:** 10.3390/biomedicines11071794

**Published:** 2023-06-23

**Authors:** Qingyu Wu, Shuo Li, Xianrui Zhang, Ningzheng Dong

**Affiliations:** 1Cyrus Tang Hematology Center, Collaborative Innovation Center of Hematology, State Key Laboratory of Radiation Medicine and Prevention, Soochow University, Suzhou 215123, China; zhangxianrui94@126.com (X.Z.); ningzhengdong@suda.edu.cn (N.D.); 2Department of Cancer Biology, Lerner Research Institute, Cleveland Clinic, Cleveland, OH 44195, USA; lis3@ccf.org; 3NHC Key Laboratory of Thrombosis and Hemostasis, Jiangsu Institute of Hematology, Soochow University, Suzhou 215006, China

**Keywords:** adipose tissue, corin, hepsin, matriptase-2, type II transmembrane serine protease, thermogenesis

## Abstract

Adipose tissue is a crucial organ in energy metabolism and thermoregulation. Adipose tissue phenotype is controlled by various signaling mechanisms under pathophysiological conditions. Type II transmembrane serine proteases (TTSPs) are a group of trypsin-like enzymes anchoring on the cell surface. These proteases act in diverse tissues to regulate physiological processes, such as food digestion, salt-water balance, iron metabolism, epithelial integrity, and auditory nerve development. More recently, several members of the TTSP family, namely, hepsin, matriptase-2, and corin, have been shown to play a role in regulating lipid metabolism, adipose tissue phenotype, and thermogenesis, via direct growth factor activation or indirect hormonal mechanisms. In mice, hepsin deficiency increases adipose browning and protects from high-fat diet-induced hyperglycemia, hyperlipidemia, and obesity. Similarly, matriptase-2 deficiency increases fat lipolysis and reduces obesity and hepatic steatosis in high-fat diet-fed mice. In contrast, corin deficiency increases white adipose weights and cell sizes, suppresses adipocyte browning and thermogenic responses, and causes cold intolerance in mice. These findings highlight an important role of TTSPs in modifying cellular phenotype and function in adipose tissue. In this review, we provide a brief description about TTSPs and discuss recent findings regarding the role of hepsin, matriptase-2, and corin in regulating adipose tissue phenotype, energy metabolism, and thermogenic responses.

## 1. Introduction

Adipose tissue is an essential organ in energy balance, metabolic homeostasis, and thermogenesis [1,2]. There are two major types of adipose tissue: white adipose tissue (WAT) and brown adipose tissue (BAT), which serve distinct functions; WAT is primarily for energy storage, whereas BAT is for thermogenesis. The cellular components and activities in adipose tissue are closely regulated by local and systemic signaling mechanisms. Abnormalities in adipose tissue function can lead to metabolic disorders, e.g., obesity and type 2 diabetes, which are major risk factors leading to cardiovascular disease.

Serine proteases are a class of proteolytic enzymes that modify protein structure and function [3]. These proteases act in a wide range of tissues to control physiological processes, including cell growth, embryonic development, and tissue homeostasis. Dysregulated serine protease activities are major contributing factors in cardiovascular disease, neuronal disease, and cancer. In adipose tissue, serine proteases (e.g., adipsin, kallikreins, and proprotein convertases) participate in the activation of growth factors, hormones, adipokines, neuropeptides, and metalloproteinases, which are of metabolic significance. Serine protease inhibitors, also known as serpins (e.g., plasminogen activator inhibitor 1 and vaspin), also play a role in regulating adipose tissue function and are implicated in obesity and metabolic dysfunction.

Type II transmembrane serine proteases (TTSPs) are a group of trypsin-like enzymes important in physiological processes [4,5,6,7]. Deficiencies in TTSPs contribute to cardiovascular disease, skin disease, hematological disease, and cancer. In recent years, several TTSPs have been identified as key regulators in adipose tissue phenotype and function. In this review, we provide a brief introduction of TTSPs and discuss the roles of hepsin, matriptase-2, and corin in adipose tissue biology and thermogenesis.

## 2. TTSPs: General Background

### 2.1. Protein Domains and Post-Translational Modifications

Proteolytic cleavage is one of the most common mechanisms in activating or degrading proteins. The cleavage can also generate protein fragments with new functions, thereby contributing to the functional diversity of the proteome. In the human genome, ~2% of genes encode proteolytic enzymes, among which serine proteases are a major class [8]. TTSPs are a family of trypsin-like serine proteases. In humans, the TTSP family has 17 members, all of which have a single-span transmembrane segment near the N-terminus and a trypsin-like serine protease domain at the C-terminus [4,5]. Based on protein modules between the transmembrane segment and the protease domain, the TTSP family can be divided into four subgroups: the human airway trypsin-like protease (HAT) subgroup, including HAT, HAT-like (HAT-L) 2-5, TMPRSS11A (transmembrane protease serine 11A), and DESC1 (differentially expressed in squamous cell carcinoma 1); the hepsin subgroup, including hepsin, TMPRSS2-5, enteropeptidase, and MSPL (mosaic serine protease large-form); the matriptase subgroup, including matriptase, matriptase-2-3, and polyserase-1; and the corin subgroup with corin only. Figure 1 shows domain combinations in selected TTSPs representing each of the four subgroups.

TTSPs are synthesized as zymogens, which are activated by proteolytic cleavage at a conserved site before the C-terminal protease domain. To date, distinct molecular and cellular mechanisms have been identified in TTSP zymogen activation. For example, some TTSPs are activated via autocatalysis either intracellularly, e.g., TMPRSS2 [9], TMPRSS11A [10], and TMPRSS13 [11], or on the cell surface, e.g., hepsin [12] and matriptase-2 [13]. Some TTSPs, e.g., enteropeptidase [14] and corin [15], are activated on the cell surface by other proteolytic enzymes. For matriptase, both autoactivation and prostasin-mediated activation have been reported [16,17,18,19,20]. Cellular responses to inflammation and oxidative stress also play a regulatory role in TTSP activation [21,22,23]. The diverse mechanisms in TTSP zymogen activation likely reflect individual protein structures and specific cellular environments in which the proteases are expressed.

N-glycosylation is a common post-translational modification important in protein folding, intracellular trafficking, and cell surface expression [24,25]. It has been shown that N-glycans at distinct sites in TTSPs interact with calnexin, an endoplasmic reticulum (ER) chaperone which facilitates glycoprotein folding, quality control, and ER exiting [26]. Abolishing selected N-glycosylation sites in TTSPs prevents the interaction with calnexin, resulting in ER retention of poorly folded proteins and triggering the unfolded protein response and ER stress [26]. Calreticulin is another ER chaperone in the calnexin–calreticulin cycle of glycoprotein folding [27,28]. To date, it remains unclear if calreticulin is involved in the folding and intracellular trafficking of TTSPs.

The transmembrane domain in TTSPs tethers the protein on the cell surface. In polarized epithelial cells, TTSPs may be targeted to a specific region of the cell surface. For example, corin is targeted to the apical, but not basolateral, membrane in renal epithelial cells via a Rab11a-dependent sorting mechanism [29]. In non-polarized cardiomyocytes, no selective cell membrane distribution of corin is observed [29,30]. Once on the cell surface, TTSPs may undergo ectodomain shedding, a common process to remove cell surface proteins [31,32]. Both ectodomain shedding and inhibition by protease inhibitors are important regulatory mechanisms to reduce protease activities. In several TTSPs, including matriptase [33,34,35,36,37], matriptase-2 [38], TMPRSS13 [39], corin [40], and hepsin [12], autocatalysis and metalloproteinase or cysteine protease-mediated shedding has been reported. Moreover, endocytosis has been identified as another mechanism in regulating matriptase-2 expression and function [38,41]. It remains unclear if TTSPs undergo cell membrane recycling, which is common among many transmembrane proteins.

### 2.2. Physiological Functions

Based on studies in cells, knockout (KO) mice, and human genetics, physiological functions of many TTSPs have been reported. For example, enteropeptidase is known for trypsinogen activation in food digestion [42]. Matriptase is essential for dermal and intestinal epithelial integrity [20,43,44,45,46,47]. TMPRSS3 is required for cochlear hair cell survival and normal hearing [48,49,50]. HAT (also known as TMPRSS11D) regulates bronchial epithelial function, including cell proliferation, adhesion, and mucin production [51,52,53,54]. TMPRSS11A is important in skin wound healing [55]. TMPRSS13 and HAT-L4 are critical for skin barrier function [56,57,58]. The function of hepsin [59,60], matriptase-2 [61,62], and corin [40,63] in liver function, iron metabolism, and cardiovascular biology, respectively, will be discussed in following sections. In mice, deficiencies in Tmprss2, Tmprss11a, Tmprss11b (also known as HAT-like 5), Tmprss11c (also known as HAT-like 3), HAT, Tmprss11e (also known as DESC1), and Tmprss11g (also known as HAT-like 2) do not cause discernable abnormalities [57,64,65]. More phenotypic studies are needed to elucidate the function of those TTSPs in vivo.

In addition to their roles in normal tissues, TTSPs also act under pathological conditions. In cancers, for example, TTSPs, such as matriptase, matriptase-2, hepsin, and TMPRSS13, promote cancer progress by degrading extracellular matrix proteins and activating signaling pathways [7,59,66,67,68]. In human and mouse airways, multiple TTSPs (e.g., HAT, matriptase, TMPRSS2, TMPRSS4, TMPRSS11A, and TMPRSS13) are expressed on the epithelial surface [69,70]. Proteolytic cleavage of viral particle surface proteins by TTSPs is a key process in host cell entry for influenza A viruses and coronaviruses [69,71,72]. To date, major efforts are ongoing to develop TTSP inhibitors for blocking airway viral infection [73,74]. Hepsin-mediated cleavage STING (stimulator of interferon genes) has also been identified as an important mechanism in inhibiting type I interferon induction in viral infection [75].

## 3. TTSP Expression in Adipose Tissues

Adipose tissue is an endocrine organ secreting many adipokines and hormones of metabolic importance [2]. In adipose tissue, there are many cell populations, including adipogenic progenitors, adipocytes, fibroblasts, immune cells, and vascular cells. Multiple single-cell and spatial transcriptomic studies, together with proteomic analysis, have identified distinct gene profiles in specific cell types [76,77]. Amazingly, ~70% of protein-coding genes in the human genome are expressed in adipose tissue, an indication of dynamic biological activities [78,79]. The gene-profiling studies provide important insights into the gene and cellular landscapes of human adipose tissue.

Analysis of the Human Protein Atlas database (www.proteinatlas.org) indicates that many genes in adipose tissue encode serine proteases, metalloproteinases, and cysteine proteases. Among them, several TTSP-encoding genes, including *HPN* (encoding hepsin), *ST14* (encoding matriptase), *TMPRSS3*, *TMPRSS5*, and *TMPRSS11E*, are detected. Since all TTSPs contain a cell membrane-anchoring domain, the detected TTSPs are expected to function in adipose tissue. To date, the role of matriptase, TMPRSS3, TMPRSS5, and TMPRSS11E in adipose tissue remains to be defined. Conceivably, TTSPs expressed in non-adipose tissues may also alter adipocyte function via indirect mechanisms. In the following sections, we will discuss the role of hepsin, matriptase-2, and corin in regulating adipose tissue function and the underlying mechanisms.

## 4. Hepsin in Adipose Tissue Differentiation

### 4.1. Hepsin Protein and Function

Hepsin was discovered in human hepatoma cells [80]. Based on the cDNA sequence, hepsin was predicted to contain an N-terminal transmembrane domain and an extracellular region with a scavenger receptor cysteine-rich (SRCR) domain and a C-terminal trypsin-like protease domain (Figure 1). As shown in biochemical studies, the transmembrane domain is unnecessary for hepsin proteolytic activity. Soluble forms of hepsin are capable of cleaving protein and small peptide substrates [81]. The findings are consistent with data in other TTSPs, indicating that the primary function of the transmembrane domain is for cell surface anchoring but not catalytic activity enhancing.

Consistent with its abundant expression in the liver, the primary function of hepsin is to regulate hepatocyte morphology and function. In hepsin KO mice, hepatocytes are larger in size and sinusoidal capillaries are narrower, compared to those in wild-type (WT) mice [60,82]. Hepsin-deficient mice also have reduced levels of triglyceride and glycogen in the liver and low levels of triglyceride, cholesterol, free fatty acids, and albumin in plasma [83,84]. In humans, the *HPN* locus has been identified as a major determinant of serum albumin levels [85]. These findings indicate an important role of hepsin in regulating lipid, glucose, and protein metabolism in the liver.

The hepatic function of hepsin is mediated, at least in part, by a hepatocyte growth factor (HGF)-dependent mechanism [83]. HGF is a ligand for c-Met, a key tyrosine kinase receptor in cell differentiation [86]. HGF is synthesized as a precursor, i.e., pro-HGF, which is activated on the cell surface by proteolytic cleavage. In cell-based studies, hepsin converted pro-HGF to HGF, leading to c-Met activation [81,87]. In livers from hepsin KO mice, pro-HGF processing and phosphorylation of c-Met and its downstream effectors, e.g., Akt, Gsk3, and mTOR, were decreased [60,83]. HGF treatment in hepsin KO mice stimulated c-Met, Akt, Gsk3, and mTOR phosphorylation and glycogen production in the liver [83]. These results indicate that hepsin-mediated pro-HGF processing is critical for c-Met signaling in liver function.

In addition to the liver, hepsin is expressed in the kidney, stomach, thyroid, pancreas, inner ear, and many types of cancers. In renal tubules, hepsin-mediated cleavage is required for uromodulin polymerization in a defense mechanism to prevent bacterial infection in the urinary tract [88,89]. In the inner ear, hepsin is essential for normal cochlear development and auditory nerve function, as indicated by severe hearing loss in hepsin-deficient mice [84,90]. This function is likely mediated by hepsin-mediated pro-HGF activation since a similar hearing loss phenotype was reported in patients with *HGF* mutations [91]. In breast cancers, hepsin-mediated cleavage of fibronectin in the extracellular matrix has been identified as a key mechanism in transforming growth factor-beta signaling and cancer progression [92].

### 4.2. Role of Hepsin in Adipose Tissue Browning

As indicated in single-cell sequencing and proteomic analysis, hepsin is expressed in human adipose tissue [78,79]. Hepsin mRNA and protein were also detected in mouse WAT and BAT [83]. Compared to those in WT mice, WAT and BAT weights were lower and adipocyte sizes were smaller in hepsin KO mice, an indication of altered adiposity [83]. In molecular studies, high levels of brown adipocyte markers, such as uncoupling protein 1 (Ucp1) and cell death-inducing DFFA-like effector A (Cidea), were found in hepsin KO BAT and WAT [83]. Additionally, levels of beige adipocyte markers (Cd137 and T-box protein 1) in WAT and mitochondrial gene expression, such as *Cox7a1* (encoding cytochrome c oxidase subunit 7A1), *Cpt1b*, and *Cpt2* (encoding carnitine palmitoyltransferase 1b and 2, respectively), were increased in hepsin KO BAT and WAT [83]. These results indicate enhanced adipose tissue browning in hepsin KO mice.

Consistently, cultured interscapular BAT and inguinal WAT from hepsin KO mice were more active in glucose uptake [83], an indication of increased metabolism. Indeed, hepsin KO mice exhibited a high metabolic rate, as indicated by increased food and water intakes, core body temperature, heat generation, elevated O_2_ consumption, CO_2_ generation, and respiratory exchange ratio [83]. In contrast, no changes in motor activities were observed in hepsin KO mice. Unlike WT mice, hepsin KO mice were protected from developing hyperlipidemia and obesity on a high-fat diet. Hepsin deficiency also reduced diabetes and obesity in *db*/*db* mice [83]. These data show that hepsin is an important regulator in adipose tissue phenotype and function in mice.

Like in the liver, the hepsin function in adipose tissue is likely mediated by the HGF and c-Met pathway. Both pro-HGF and c-Met are expressed in adipose tissue [83]. In hepsin KO mice, c-Met phosphorylation in adipose tissue was decreased [83]. Moreover, Akt phosphorylation, which suppresses Ucp1 expression in brown adipocytes in a Pgc-1α-mediated mechanism [93], was also reduced in hepsin KO BAT. In culture, differentiated brown adipocytes from hepsin KO mice had high levels of Pgc-1α and Ucp1 expression and O_2_ consumption [83]. HGF treatment of adipocytes from hepsin KO mice lowered Ucp1 expression in culture [83]. These data point to a cell-autonomous hepsin function in adipose tissue to suppress adipocyte browning via c-Met signaling (Figure 2).

### 4.3. Regulation of Hepsin Expression in Adipose Tissue

In prostate cancer, *HPN* is one of the most highly upregulated genes. Similar *HPN* upregulation occurs in breast, ovarian, liver, and stomach cancers [7]. Oncogenic Ras has been identified as a stimulator in hepsin expression via Raf-MEK-ERK signaling [94]. In normal tissues, the regulation of *HPN* expression is poorly understood. In WT mice on a high-fat diet, no *Hpn* expression changes were observed in the liver or adipose tissue [83]. In contrast, reduced *Hpn* mRNA levels were found in the liver and adipose tissue in *db*/*db* mice [83]. In WT mice exposed to cold (4 °C), elevated *Hpn* expression was noticed in the liver and adipose tissue [83]. These findings indicate that genetic and environmental factors may play a role in regulating hepsin function. Further studies will be important to determine if the observed changes are due to direct effects on *Hpn* expression or secondary effects in response to metabolic imbalance or altered body temperature.

## 5. Matriptase-2 in Iron Metabolism and Adiposity

### 5.1. Matriptase-2 in Iron Metabolism

Matriptase-2 is a hepatic TTSP encoded by the *TMPRSS6* gene in humans [95]. The protein consists of an N-terminal cytoplasmic segment, a transmembrane domain, and an extracellular region with a SEA (sea urchin sperm protein, enteropeptidase, and agrin) domain, two CUB (complement factor C1s/C1r, urchin embryonic growth factor, and bone morphogenetic protein) domains, three low-density lipoprotein receptor (LDLR) class-A repeats, and a C-terminal trypsin-like protease domain [95] (Figure 1). In hepatocytes, matriptase-2 is synthesized as a zymogen, which is activated on the cell surface by autocatalysis [13]. The primary function of matriptase-2 is to regulate iron metabolism by suppressing hepcidin, a hormone that downregulates the iron transporter ferroportin [61,95]. In humans and mice, matriptase-2 deficiency results in high levels of serum hepcidin and low levels of ferroportin, which prevents iron efflux in macrophages, intestinal epithelial cells, and hepatocytes, thereby causing iron deficiency anemia [96,97].

In humans, hepcidin is encoded by the *HAMP* gene, which is upregulated by bone morphogenetic protein (BMP) signaling [98]. In hepatocytes, the binding of BMP to its cell surface receptors activates downstream signaling, thereby increasing *HAMP* expression [99]. Hemojuvelin, a co-receptor for BMP in *HAMP* induction, was identified as a main target of matriptase-2 in hepatocytes [100]. Proteolytic cleavage of hemojuvelin by matriptase-2 reduces BMP signaling and hence hepcidin expression. More recent studies show that hemojuvelin is not the only substrate of matriptase-2. In human hepatoma cells and mouse models, matriptase-2 also cleaves other BMP receptors on the cell surface, including BMP receptor type 1A (also known as ALK3), activin receptor IIA (also known as ActRIIA), hemochromatosis protein, and transferrin receptor-2 that are important in the hepcidin induction pathway [62,101].

The function of most proteases depends on their proteolytic activity. Likewise, the matriptase-2 function in inhibiting BMP signaling was also expected to be mediated by its proteolytic activity. Surprisingly, matriptase-2 binding, via its CUB and protease domains, appears sufficient to sequester and decrease its substrates on the cell surface, resulting in reduced hepcidin expression [101]. These findings show that matriptase-2 may suppress hepcidin expression by targeting multiple proteins in the BMP signaling pathway via proteolytic and non-proteolytic mechanisms [101]. In this regard, matriptase and corin in the TTSP family also exhibit non-proteolytic functions [102,103]. Currently, matriptase-2 inhibitors are being developed to treat iron overload diseases, such as hemochromatosis [104]. Most approaches aim to block matriptase-2 catalytic activity. The finding of the non-proteolytic function in matriptase-2 suggests that targeting matriptase-2 binding to its substrates may be considered as a new approach to increase hepcidin expression as a therapy for iron overload diseases [101].

### 5.2. Matriptase-2 in Lipolysis and Obesity

Dysregulated iron metabolism and hepcidin expression are associated with obesity, type 2 diabetes, and insulin resistance. For example, hypoferremia, a deficiency of iron in the circulating blood, occurs more frequently in obese individuals [105]. High levels of serum hepcidin and poor iron absorption are also common in obese children and women [106,107]. In obese patients, hepcidin expression is increased in visceral and subcutaneous adipose tissue where levels of inflammatory proteins, e.g., IL-6 and C-reactive protein, are high [108]. In mice, dietary iron overload increases serum hepcidin levels, leading to insulin resistance and decreased visceral adipose tissue [109]. These results suggest a regulatory function of hepcidin in adipose tissue metabolism and/or inflammation.

Given its function in inhibiting hepcidin expression, matriptase-2 may also play a role in regulating metabolism in adipose tissue. In supporting this hypothesis, Tmprss6 KO mice, which had high levels of hepcidin and low levels of plasma iron, were resistant to high-fat diet-induced obesity and hepatic steatosis, as indicated by less total body fat mass, reduced adipocyte sizes and weights in WAT and BAT, and diminished lipid deposits and triglyceride contents in the liver [110]. Additionally, high-fat diet-treated Tmprss6 KO mice exhibited improved glucose tolerance and insulin sensitivity and enhanced fat lipolysis with increased adipose triglyceride lipase expression and phosphorylation/activation of hormone-sensitive lipase in WAT [110].

The mechanism underlying the protective role of matriptase-2 deficiency against obesity and hepatic steatosis is not fully understood. In principle, the observed resistance to obesity in high-fat diet-treated Tmprss6 KO mice could be due to high hepcidin levels or low iron levels or both. When hypoferremia—but not high hepcidin levels—in Tmprss6 mice was corrected via iron injection, the protection against obesity remained [110]. Moreover, iron-treated WT mice, in which hepcidin levels were high, were also resistant to high-fat diet-induced obesity and hepatic steatosis, suggesting that a high hepcidin level is primarily responsible for the obesity-resistant phenotype [110] (Figure 3). In supporting this idea, injection of an anti-hemojuvelin neutralizing antibody in Tmprss6 KO mice reduced hepcidin levels and eliminated the obesity-resistant phenotype [110]. These findings underscore the regulatory role of matriptase-2 and hepcidin in lipid and adipose metabolism. Currently, there is no evidence of matriptase-2 expression in adipose tissue. Additional studies will be important to understand how elevated hepcidin levels increase lipolysis in adipose tissue in Tmprss6 KO mice and if matriptase-2 deficiency also protects against obesity in humans.

## 6. Corin in Adipose Tissue Phenotype and Thermogenesis

### 6.1. Corin in Pro-ANP Processing

Corin was discovered in the heart [40]. The primary function of corin is to activate atrial natriuretic peptide (ANP), a pleiotropic hormone produced in the heart and, at low levels, in non-cardiac tissues. In the heart, ANP serves as a key component in a hormonal mechanism to control blood volume and pressure by relaxing vessels and increasing renal sodium and water excretion [111]. Moreover, ANP regulates cardiac morphology, function, and aging [112,113]. In the pregnant uterus, ANP promotes decidualization and spiral artery remodeling in an auto/paracrine mechanism critical for fetal growth [114]. Impaired ANP expression and/or function have been implicated in major cardiovascular diseases, including atrial fibrillation, hypertension, cardiac hypertrophy, heart failure, and pre-eclampsia [111,115].

Like many peptide hormones, ANP is derived from its precursor, i.e., pro-ANP, via proteolytic cleavage. The cleavage is carried out by corin on the cell surface [40]. Like other TTSPs, corin has a single-pass transmembrane domain near the N-terminus. In the extracellular region, there are two frizzled domains, eight LDLR class-A repeats, an SRCR domain, and a C-terminal protease domain (Figure 1). Corin-mediated pro-ANP cleavage occurs not only in the heart, but also in non-cardiac tissues, including the kidney, skin, and uterus [114,116,117]. Disruption of the *Corin* gene prevents pro-ANP processing in mice, causing salt-sensitive hypertension that is also observed in ANP KO mice. In humans, dysfunctional *CORIN* variants are associated with impaired pro-ANP processing and hypertensive diseases [40,118,119,120,121].

### 6.2. Role of ANP in Lipid Metabolism in Adipose Tissue

In addition to its function in salt–water balance and blood pressure, ANP enhances cellular activity and lipid metabolism in adipose tissue [122]. In cultured adipocytes, ANP treatment increases lipolysis by stimulating phosphorylation of the hormone-sensitive lipase and perilipin A, a lipid droplet coating protein that recruits lipases in adipocytes [123]. This ANP function is mediated by increasing intracellular cGMP generation and subsequent activation of the cGMP-dependent protein kinase I (cGKI). Inhibition of cGKI, but not the cAMP-dependent protein kinase (PKA), reduced ANP-mediated lipolysis in adipocytes [123]. Consistently, intravenous ANP infusion in humans increased lipid hydrolysis and oxidation in subcutaneous abdominal adipose tissue [124]. Similarly, enhancing natriuretic peptide signaling by deleting the *Nprc* gene (encoding natriuretic peptide clearance receptor) in mouse adipose tissue decreased adipocyte size, stimulated lipid metabolism, and prevented diet-induced obesity and insulin resistance [125]. Conversely, in mouse and rat models of obesity, cardiac ANP expression is suppressed [126,127]. It appears that the ANP-mediated lipid-mobilizing function is independent of sympathetic nervous system activation [128]. In addition to adipose tissue, ANP also promotes lipid oxidation in skeletal muscles [129]. These data indicate that ANP is an important hormone that connects the heart and adipose tissue in controlling lipid and energy metabolism.

### 6.3. Role of ANP in Adipose Tissue Browning and Thermogenesis

BAT is essential for non-shivering thermogenesis, a survival response in cold environments [1]. ANP has been shown to promote the browning thermogenic program in human and mouse adipocytes [130]. In ANP-treated adipocytes, levels of UCP1 and mitochondrial gene expression are markedly increased. This function is mediated by a signaling pathway involving protein kinase G (PKG) and p38 mitogen-activated protein kinase (MAPK) [130,131], resulting in the activation of the transcriptional co-activator PGC-1α, a key regulator of mitochondrial biogenesis and thermogenesis in BAT [93]. In mice, ANP administration enhanced adipose tissue browning and thermogenetic activities and reduced insulin resistance induced by a high-fat diet [132]. Likewise, treatment of B-type natriuretic peptide, another natriuretic peptide with similar activities to those of ANP, elevated Ucp1 expression in adipocytes in culture and in mice [130,133]. Consistently, ANP deficiency reduces BAT activity, resulting in cold intolerance in mice [134]. Based on comparative biology studies, the natriuretic peptides were originated in primitive fish species to regulate electrolyte homeostasis [135,136]. Apparently, the function of ANP has expanded in endothermic animals to provide an additional adaptive advantage in new environments where salt intake and temperatures fluctuate widely.

### 6.4. Role of ANP in Adipose Tissue Inflammation

In obese individuals, chronic inflammation in adipose tissue contributes to insulin resistance, metabolic dysfunction, and risks of cardiovascular disease [137,138]. Inhibition of inflammation has been considered as a therapeutic target in obesity. ANP is anti-inflammatory [139,140]. In cultured human subcutaneous adipose tissue, ANP treatment inhibited the expression and secretion of multiple adipokines (e.g., leptin and retinol-binding protein-4) in adipocytes and cytokines (e.g., TNFα, IL-6, and monocyte chemoattractant protein-1) in macrophages [141], which are implicated in inflammation and metabolic dysfunction. In mice, inhibition of natriuretic peptide removal by disrupting the *Nprc* gene decreased inflammation and oxidative stress in adipose tissue, particularly when the mice were in a pro-atherosclerotic ApoE^−/−^ background [142]. Human genetic studies have also linked *NPRC* alleles to risks of abdominal adiposity [143]. These findings underscore the role of the ANP-dependent mechanism in preventing chronic inflammation in adipose tissue, which contributes to insulin resistance and type 2 diabetes.

### 6.5. Impaired Adipose Tissue Browning and Thermogenesis in Corin KO Mice

Corin is essential for ANP activation. Given the role of ANP in adipose tissue, it is anticipated that corin deficiency may alter adipose tissue phenotype. Indeed, a recent study reported that Corin KO mice had impaired adipocyte browning, as observed in ANP KO mice [144]. Particularly, Corin KO mice had an increased fat mass/body weight ratio, larger adipocyte sizes in WAT, and reduced Ucp1 and Cidea expression in BAT. Moreover, decreased p38 Mapk phosphorylation, Pgc-1α levels, and mitochondrial gene expression were also observed in BAT from Corin KO mice [144]. In culture, ANP treatment increased Ucp1 expression in BAT-derived adipocytes from Corin KO mice [144]. These data indicate that corin deficiency prevents ANP generation, thereby impairing cGKI/hormone-sensitive lipase-mediated lipolysis and the p38 Mapk-Pgc-1α-Ucp1 signaling pathway in adipose tissue browning in mice (Figure 4).

Consistent with impaired adipocyte browning, Corin KO mice had a phenotype of poor thermogenic response and cold intolerance [144]. When tested at regular room temperature (~22 °C), Corin KO mice exhibited no major metabolic abnormalities, as measured by motor activity, heat generation, O_2_ consumption, CO_2_ production, water and food intakes, and core body temperature. Upon cold exposure (4 °C), core body temperature deceased more rapidly in Corin KO mice than in WT mice, indicating that Corin KO mice are prone to developing hypothermia [144]. In WT mice exposed to cold, blood glucose level and Ucp1 expression were increased in WAT and BAT mice, as part of the thermogenic response. In contrast, such a response was markedly diminished in cold-exposed Corin KO mice [144]. These results indicate that corin-mediated ANP activation is critical for non-shivering thermogenesis in response to cold exposure in mice. In humans, CORIN and NPPA (encoding prepro-ANP) expression was found in visceral and subcutaneous adipose tissue from obese individuals [145], suggesting the potential auto- or paracrine mechanism of corin and ANP in adipose tissue. In C57BL/6J WT mice, however, Corin and Nppa mRNAs were not detected via RT-PCR in inguinal WAT and interscapular BAT [144]. Further studies are required to understand if the corin and ANP function in adipose tissue is mediated by endocrine and/or auto/paracrine mechanism(s) in humans and mice.

It is worth noting that sweating is a key thermoregulatory mechanism to reduce core body temperature in hot environments or during hard labor or exercises. Among all mammals, humans have the most abundant eccrine sweat glands present, covering nearly the entire skin surface [146]. This anatomical characteristic offers an advantage for humans to live in hot and dry territories [147,148]. As reported recently, both corin and ANP are expressed in human and mouse skin eccrine sweat glands [117]. In mice, corin deficiency reduced sodium and sweat excretion in paw skin [117]. Apparently, the corin function, via ANP activation serves as an anti-aldosterone mechanism to promote sodium and water excretion in the sweating duct [117]. These findings highlight the role of corin in thermoregulation via two independent mechanisms: enhancing adipocyte browning and non-shivering thermogenesis under cold conditions and promoting sweat excretion in skin eccrine glands under hot conditions. More studies will be important to determine if and how *CORIN* variants may alter adipocyte function, thermoregulation, and adaptive fitness in humans.

## 7. Conclusions and Perspectives

It has been more than 20 years since TTSPs were cloned and recognized as a distinct group of proteolytic enzymes [149,150]. The importance of TTSPs in many physiological processes, such as sodium homeostasis, iron metabolism, epithelial integrity, vascular remodeling, and auditory nerve development, is well documented. Defects in TTSPs have been shown to cause cardiovascular, hematological, intestinal, skin, and hearing disorders. Upregulated TTSP expression also contributes to cancer development and progression. In respiratory viral infection, epithelial TTSPs in human airways play a major role in activating viral particle proteins, thereby enhancing viral infectivity.

Adipose tissue is essential in energy metabolism and thermoregulation. Dysregulated adipose function is a major factor in the pathogenesis of metabolic and cardiovascular diseases. Recent studies in mouse models have shown that members of the TTSP family, namely hepsin, matriptase-2, and corin, play an important role in regulating adipose phenotype and thermogenesis via direct signaling or indirect hormonal mechanisms. More studies are expected to verify these findings and to examine if additional underling mechanisms are involved. It will also be important to determine if hepsin, matriptase-2, and corin play similar roles in human adipose tissue and participate in metabolic diseases, such as obesity, dyslipidemia, and type 2 diabetes.

As revealed via single-cell and spatial transcriptomic studies, there are additional TTSPs (e.g., matriptase, TMPRSS3, TMPRSS5, and TMPRSS11E) expressed in human adipose tissue. Further studies are required to determine if these TTSPs act in adipose tissue to regulate cellular components and activities. Findings from such studies shall help to understand the role of TTSPs in adipose tissue function, metabolic homeostasis, and thermogenesis, which are of great biological significance.

## Figures and Tables

**Figure 1 biomedicines-11-01794-f001:**
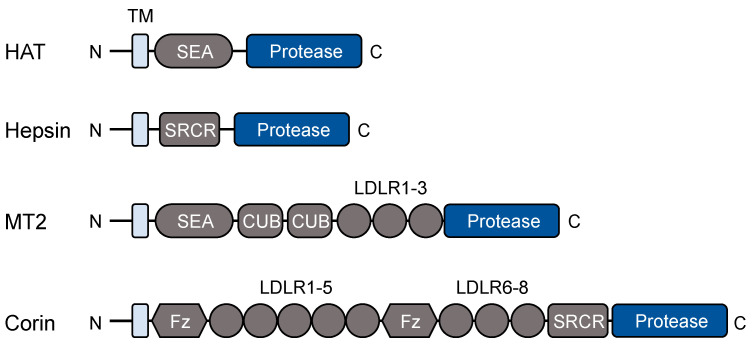
Protein modules in selected TTSPs. All TTSPs contain an N-terminal transmembrane (TM) domain and a C-terminal trypsin-like protease domain. Protein modules between the transmembrane domain and the protease domain vary in individual TTSPs. HAT, human airway trypsin-like protease; SEA, sea urchin sperm protein, enteropeptidase, and agrin; SRCR, scavenger receptor cysteine-rich; MT2, matriptase-2; CUB, complement factor C1s/C1r, urchin embryonic growth factor, and bone morphogenetic protein; LDLR, low-density lipoprotein receptor; Fz, frizzled.

**Figure 2 biomedicines-11-01794-f002:**
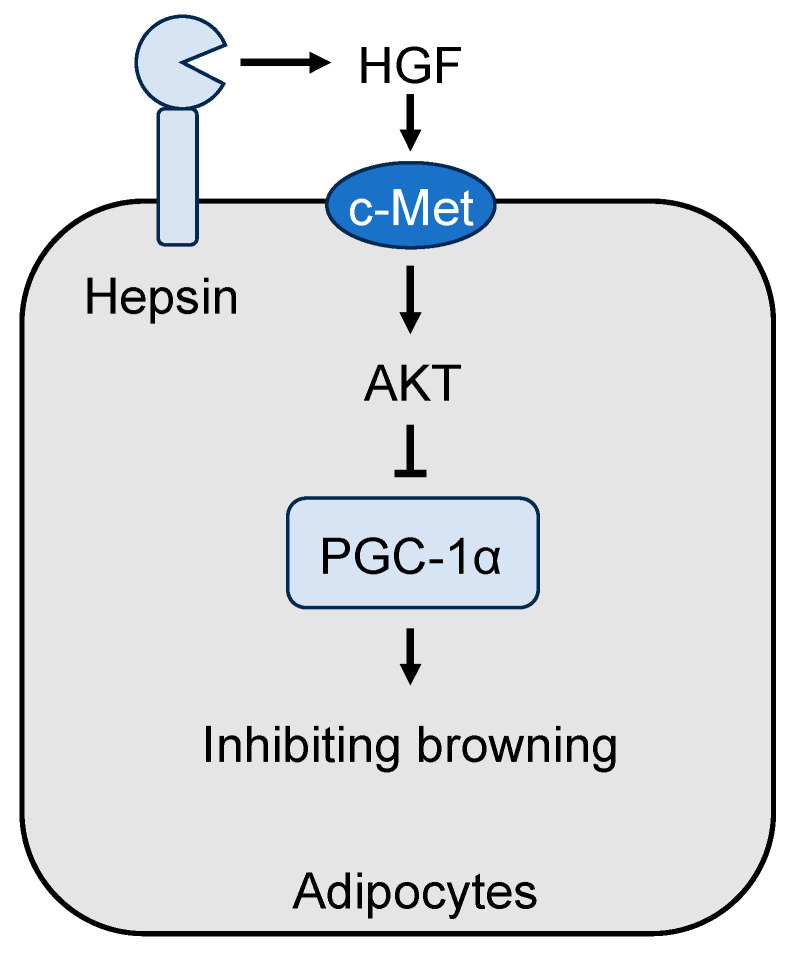
A model of the hepsin/HGF/c-Met pathway in adipocytes. Hepsin activates HGF (hepatocyte growth factor), which in turn activates c-Met signaling and the downstream effectors including AKT (also known as protein kinase B). Subsequently, AKT inhibits PGC-1α, a transcriptional co-activator in adipocyte browning and mitochondrial gene expression.

**Figure 3 biomedicines-11-01794-f003:**
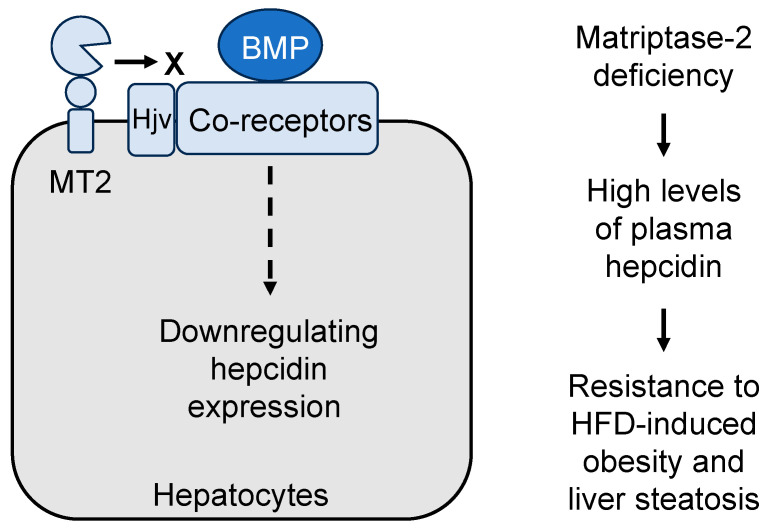
Matriptase-2 function in hepcidin expression. In hepatocytes, bone morphogenetic protein (BMP) signaling enhances hepcidin expression. Matriptase-2 (MT2) sequesters and cleaves hemojuvelin (Hjv) and other BMP co-receptors on the cell surface, thereby downregulating BMP signaling and hepcidin expression. Matriptase-2 deficiency increases hepcidin levels in plasma, resulting in a protective phenotype against high-fat diet (HFD)-induced obesity and liver steatosis in mice.

**Figure 4 biomedicines-11-01794-f004:**
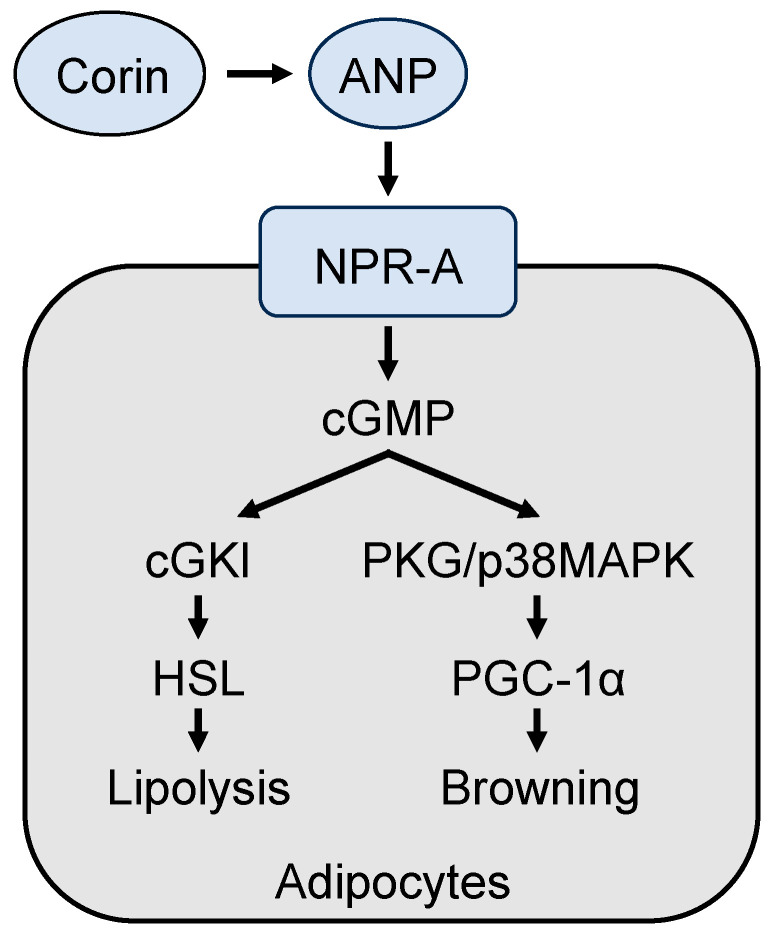
A model of corin and ANP-mediated mechanisms in promoting lipolysis and browning in adipocytes. Corin activates ANP, which in turn binds to and activates its receptor, natriuretic peptide receptor A (NPR-A), thereby stimulating intracellular cGMP production. Increased cGMP levels promote lipolysis and adipocyte browning via the cGMP-dependent protein kinase I (cGKI)/hormone-sensitive lipase (HSL) pathway and the protein kinase G (PKG)/p38 mitogen-activated protein kinase (MAPK)/PGC-1α pathway, respectively.

## Data Availability

Not applicable.

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
