# Peer review of "Type II Transmembrane Serine Proteases as Modulators in Adipose Tissue Phenotype and Function"

_biomedicines, 2023, doi:10.3390/biomedicines11071794_

Round 1

Reviewer 1 Report

The manuscript biomedicines-2440429, entitled Type II Transmembrane Serine Proteases as Modulators in Adipose Tissue Phenotype and Function by Qingyu Wu  reviewed the type II transmembrane serine proteases (TTSPs), a group of trypsin-like enzymes anchoring on the cell surface. These proteases act in diverse tissues to regulate physiological processes, such as food digestion, salt-water balance, iron metabolism, epithelial integrity, and auditory nerve development. Several members of the TTSP family, namely, hepsin, matriptase-2, and corin, have been shown to play a role in regulating lipid metabolism, adipose tissue phenotype, and thermogenesis, via direct growth factor activation or indirect hormonal mechanisms. In mice, hepsin deficiency increases adipose browning and protects from high-fat diet-induced hyperglycemia, hyperlipidemia, and obesity. Similarly, matriptase-2 deficiency increases fat lipolysis and reduces obesity and hepatic steatosis in high-fat diet-fed mice.

The review work is well documented.

The revision of literature deep.

The text is well written, clear and informative.

Figure are informative and of excellent quality.

Minor notes:

Line340: the title should be with the text in the following page.

Line 389: the title should be with the text in the following page.

Enlish quality is reather high

Author Response

Reviewer's Minor notes:

Line340: the title should be with the text in the following page.

Line 389: the title should be with the text in the following page.

Response: We thank the reviewer for the positive feedback. The section title lines now are with the sections.

Reviewer 2 Report

The review entitled “Type II Transmembrane Serine Proteases as Modulators in Adipose Tissue Phenotype and Function” by Wu Q. et al. gives an overview of the state of the art related to the role of the Type II transmembrane Serine proteases in adipose tissue phenotype and function changing, in particular of hepsin, matriptase-2, and corin. The adipose tissue is essential in the regulation of thermogenesis (brown adipose tissue) and energy storage (white adipose tissue) and it is regulated by local and systemic signaling mechanisms. The altered function in the adipose tissue can lead to metabolic disorders, such as type 2 diabetes and obesity, which are the main risk factors for cardiovascular diseases. Serine proteases act by modifying the structure of proteins (promoting an activation or deactivation), regulating many important processes, such as embryonic development, tissue homeostasis, and cell growth, and an alteration or dysfunction of their activity is correlated with cancer, cardiovascular and neuronal diseases. In particular, in adipose tissue, serine proteases activate many effectors of the metabolic route for example hormones, growth factors, adipokines, metalloproteinases, and neuropeptides. The use of inhibitors of serine proteases (serpins) is implicated in metabolic dysfunction and obesity. In particular, recent studies identified Type II transmembrane serine proteases (TTSPs) as a group of trypsin-like enzymes (family composed of four subgroups) implicated in cardiovascular, hematological, skin diseases and cancer and the regulation of adipose tissue phenotype and function. Among these enzymes, this review is focused on the role of hepsin, matriptase-2, and corin in adipose tissue. In particular, the synthesized zymogens of hepsin and matriptase-2 are triggered by intracellular autocatalysis, while the corin zymogen is activated by proteolysis carried out by other enzymes on the cell surface. It also reported that the matriptase-2 zymogens are activated by both autocatalysis and proteolysis prostasin-mediated. The differences in the zymogen activations of type II transmembrane serine proteases are strictly dependent on specific cellular environments. Other studies highlighted that the abolishment of some post-translational modifications (for example N-glycosylation) on Type II transmembrane serine proteases prevents the interaction with endoplasmic reticulum chaperone proteins resulting in the release of incorrectly folded proteins. Hepsin and matriptase-2 seem to stimulate progression, invasiveness, and malignancies in cancer (breast, prostate, ovarian, and stomach cancers) due to the activation of metalloproteinases that degrade the extracellular matrix. 1) Hepsine is abundant in the liver and seems to be important in the regulation of hepatocyte morphology and function. Its activity in the liver (and in the adipose tissue) is regulated by the hepatocyte growth factor hormone and in humans is important to regulate the plasmatic level of serum albumin. Hepsin prevents bacterial infections in the urinary tract and helps in normal cochlear development and the function of the auditory nerve in the inner ear. The absence of hepsine in mice knocked out resulted in altered adiposity (lower white and brown adipose tissues and smaller adipocytes) and dysfunction of the metabolism and thermoregulation. 2) In the case of matriptase-2, this Type II transmembrane serine protease is important in the regulation of the metabolism of iron maintaining the expression of ferroportin on the intestinal membrane, macrophages, and hepatocytes, thanks to the suppression of the hormone hepcidin activity that downregulates these transporters, causing anemia. Hypoferremia is frequent in obese individuals. 3) Instead, corin is identified in the heart and activates the atrial natriuretic peptide (ANP) in this organ and non-cardiac tissues (kidney, skin, and uterus) by the precursor pro-ANP. The atrial natriuretic peptide is the key molecule to controlling blood volume and pressure, especially cardiac morphology, function and aging, and artery remodeling in pregnancy. Impairment in the atrial natriuretic peptide levels (also due to disruption of the corin gene) led to cardiovascular problems (hypertension, cardiac hypertrophy, atrial fibrillation, heart failure, and preeclampsia) of which obesity is the main risk factor. Moreover, enhanced cellular activity, lipid metabolism in adipose tissue, and browning thermogenic program in humans stimulated by atrial natriuretic peptide were highlighted. To date, further studies are required to surely determine the involvement and the role of these Type II transmembrane serine proteases in adipose tissue which seems to be of great biological significance. This interesting review is well-organized in sections, well-written, and of great interest. Among the reported 159 references, about 120 are in the range of 2010-2023 including very recent studies. I suggest this review for publication.

Author Response

We thank the reviewer for the positive feedback and encouragement.

Reviewer 3 Report

In this review, Wu et al. describe type II transmembrane serine proteases (TTSP) and detail recent discoveries regarding the role of hepsin, matriptase-2, and corin in the regulation of adipose tissue phenotype, energy metabolism, and thermogenic responses. It is of interest not only to researchers working on adipose tissue and the proteases involved, but also to researchers in related fields. Therefore, I consider this review suitable for publication in Biomedicines.

Author Response

(The authors gave the same response as above.)
